# A Cognitive Sample Consensus Method for the Stitching of Drone-Based Aerial Images Supported by a Generative Adversarial Network for False Positive Reduction

**DOI:** 10.3390/s22072474

**Published:** 2022-03-23

**Authors:** Jeong-Kweon Seo

**Affiliations:** Institute of Data Science, Korea University, 145 Anam-ro, Seongbuk-gu, Seoul 02841, Korea; seojksc@korea.ac.kr; Tel.: +82-02-3290-4645

**Keywords:** scale-invariant feature transform, random sample consensus method, rigid transformation, drone-based aerial images, generative adversarial network

## Abstract

When using drone-based aerial images for panoramic image generation, the unstableness of the shooting angle often deteriorates the quality of the resulting image. To prevent these polluting effects from affecting the stitching process, this study proposes deep learning-based outlier rejection schemes that apply the architecture of the generative adversarial network (GAN) to reduce the falsely estimated hypothesis relating to a transform produced by a given baseline method, such as the random sample consensus method (RANSAC). To organize the training dataset, we obtain rigid transforms to resample the images via the operation of RANSAC for the correspondences produced by the scale-invariant feature transform descriptors. In the proposed method, the discriminator of GAN makes a pre-judgment of whether the estimated target hypothesis sample produced by RANSAC is true or false, and it recalls the generator to confirm the authenticity of the discriminator’s inference by comparing the differences between the generated samples and the target sample. We have tested the proposed method for drone-based aerial images and some miscellaneous images. The proposed method has been shown to have relatively stable and good performances even in receiver-operated tough conditions.

## 1. Introduction

To solve the problem of correspondence so that they can analyze images that share common features or objects, researchers in the field of remote sensing develop local key points, descriptions of their features, and correspondence relating to the local points [1,2]. The problem of correspondence is highly relevant in that it can be related to the registration or stitching of images [3,4], object recognition [5,6,7], object tracking [8], stereo vision [9], and so on.

Recently, as advancements in artificial intelligence (AI) have led to the development of state-of-the-art techniques around the tasks of recognition, classification, and inference, developers have actively applied the techniques of machine learning in computer vision. For example, Shan et al. [10] extracted local binary patterns and applied a support vector machine (SVM) to classify facial expressions. In other studies, Kumar et al. [11] built the gist feature [12] and used an SVM to develop an automatic plant species identification program. In object tracking, Weinzaepfel et al. [13] developed a moving quadrant scale-invariant feature transform (SIFT) [14] feature to enhance the quality of the matching images wherein they applied the idea of deep convolutional nets. In the context of applying deep learning for the correspondence problem, researchers have developed technologies along two tracks: one track has involved the development of deep learning models that operate in an end-to-end fashion, while the other track has involved the development of feature extraction models that apply existing tools to the correspondence problem. In 2016, Detone et al. [15] proposed HomographyNet; in a direct approach using an end-to-end fashion deep learning model, the authors trained a network to estimate the homography by relating the query to the target images with the labeled dataset. In 2017, Rocco et al. [16] used the layers of the VGG-16 network [17] to acquire features, and thus proposed a neural network consisting of a matching network and a regression network that mimics classical approaches to fit the affine map. In 2018, Nguyen et al. [18] proposed an unsupervised learning method to generate homography; for the experimental verification, they compared their model with SIFT-based homography using the random sample consensus method (RANSAC) [19] with the threshold set at five pixels. Meanwhile, in the other branch that applies existing feature extraction methods, in 2014, Fischer et al. [20] used the features obtained from the layers of trained convolutional neural networks (CNN) and compared them to SIFT using the descriptor matching problem. In 2019, Rodríguez et al. [21] proposed a CNN-driven patch descriptor that captures affine invariance that is based on the first stages of SIFT. In 2021, Vidhyalakshmi et al. [22] combined the Root SIFT descriptor and CNN feature to enhance the matching performance for the person re-identification problem.

Deep learning-based methods are advantageous over traditional methods for solving geometric problems due to their robustness to sparse scene structures, illumination variations, and noise; specifically, feature-based methods provide good performance when there are sufficient reliable features, but these methods often fail in low-texture environments where unique features cannot be sufficiently detected and matched [23]. Although deep learning-based methods work reliably, they do not always provide the best performance, and this shortcoming is particularly caused by the fact that there may be a lack of enough datasets that are suitable for training [18,21,24,25].

On the other hand, although the existing deep learning techniques work well for specific datasets or ad hoc problems, they do not consider directly bridging other methodologies to enhance the performance by applying their mutual and relative advantages.

In this paper, as another methodology applying deep learning, we propose using deep learning to bridge the strengths of the existing methodologies so that we can leverage the advantages of the different existing methodologies and help reduce the cost of developing new methods. From that specific perspective, our proposed method calculates the false positiveness for instances of the given methodology to determine the true transformation in the matching process of the stitching problem and to recommend the reconsideration of better methods; i.e., by calculating the false positiveness, it bridges to another method that ensures better performance, even though there are some trade-offs.

For example, we apply deep learning to verify RANSAC, a method widely used to find the true inlier of the mapping function, and we improve its performance by linking it with other a posteriori outlier rejection methods, such as optimal choice of initial correspondence inliers (OCICI) [4,26]. In the process of determining the hypothesis found through RANSAC, its authenticity is determined by training a generative adversarial network (GAN) [27]. After training various types of existing affine maps delivered by RANSAC, we first make the discriminator of the GAN determine the authenticity, after which we compare it with the affine map generated by the generator to make a final decision.

To implement the proposed method for experimental verification, we apply it to drone-based aerial images. Previous studies have shown that it is relatively difficult to solve the correspondence problem for aerial images based on drones [4,26], and our proposed method shows a way to improve the efficiency of the existing methodologies supported by deep learning technology. Actually, our proposed deep learning methodology deals with the topography of the estimated resampling operators, such as affine maps and not the images; hence, from the method’s inherent properties, it may be applicable for correspondence problems with various types of images. Our proposed method can be used regardless of the type of dataset—i.e., the trained neural network system can fit any domains of data in the real-world datasets—because the shapes of the mapping operators are not sensitive to the kinds of features in the images, as they are only influenced by the viewing angles of the objects and figures. The proposed method can act as a discriminator to eliminate outliers and bridge several feature-based matching methods to maintain the performance of stitching in a stable and efficient manner. Our experimental results are mainly focused on the drone-based images which showcases the application of the proposed method. The experimental outputs for some images of general views are miscellaneous and support a ‘ripple effect’ of the proposed method for its extensive use for general images.

In Section 2, we introduce the scheme of our proposed method. In Section 3, we present the experimental results and discuss some properties and issues related to the proposed method to verify its utility. In Section 4, we conclude the study.

## 2. Method Description

### 2.1. Extraction of Local Descriptors and Geometric Correspondence

The general approach for stitching images involves identifying key points and comparing their local descriptors to determine the transform of the key points’ geometric correspondence to generate the resampling rule of the sensed images [1,2]. In our study, we use SIFT as the descriptor, which is applied in many studies for its well-known geometric invariance properties [26,28]. Figure 1 shows the reference image and sensed image with the marks of the SIFT points, where the images are resized into a 200 × 250 resolution; we present 50 random SIFT points and the lattices of the orientations.

The descriptors in both images are then matched to form a set of correspondences that estimates the parameters of the resampling rigid transform (an affine map made of a 3 × 3 matrix [4,26]) using RANSAC.

### 2.2. Outlier Discrimination Network

While focusing on the estimation of the resampling transform, our proposed model detects the falsely estimated transforms that have been fed from the baseline method of RANSAC; we apply a 3-pixel threshold for the error tolerance of RANSAC across all of our experiments. Figure 2 presents the outlier discrimination network (ODNet) used to detect the false estimations: (i) RANSAC estimates the baseline hypothesis of the affine map (the resampling transform), which is represented in the form of 3 × 3 images in Figure 2, (ii) the result is then passed as the input into the discriminator of the pre-trained GAN (D-GAN), and (iii) through the discriminator’s judgment of falseness, a better promising method is recalled to robustly calculate the estimation (here, we recall OCICI [4]) or (iv) we confirm the result of RANSAC when it is discriminated as true.

For the test correspondence problem, we use the affine transform to resample the images. During the RANSAC process used to estimate the hypothesis, we obtain several estimated transforms that yield values of the cosine distance ranging ~>0.9995 from the best one. The measure of the cosine distance C is defined as
(1)$$C\left(I_{1},I_{2}\right)=\frac{I_{1}\cdot I_{2}}{\left|\left|I_{1}\right|\right|\left|\left|I_{2}\right|\right|}$$
where $I_{1}$ and $I_{2}$ are the vector alignments representing the parameter values of the given two transforms. We then prepare the training dataset for the discriminator of GAN. Among the total 63 pairs of consecutive drone-based aerial images taken of photovoltaic panels [29], we set aside five pairs of images that do not produce enough descriptors or for which RANSAC fails to find novel transforms, as reported in [4], and some of them are included in the test data. From the remaining 58 pairs, we organize training data with 16,378 pre-estimated transforms, including the best ones; here, for the set of tentative correspondences of the descriptors that are used to estimate the transforms using RANSAC, we apply the matching strategy of the nearest rate of distance as used in [14]. Let us suppose that ${\tilde{y}}_{1}$ and ${\tilde{y}}_{2}$ are, respectively, the nearest and second nearest descriptors in the sensed image from given descriptor $\tilde{x}$ in the reference image; then, we include ${\tilde{y}}_{1}$ into the tentative set of correspondences if it holds that
(2)$$\left|\tilde{x}-{\tilde{y}}_{2}\right|/\left|\tilde{x}-{\tilde{y}}_{1}\right|>\epsilon$$

Note that we apply the threshold parameters ϵ=1.1 and 1.2 in the process to form the training data. Figure 3 illustrates the process of training data acquisition, while Figure 4 presents the related example images overlapped by some given transforms with cosine distances ranging over several values that were calculated during the process of training the data acquisition using RANSAC.

Now, to train the GAN, we model the architecture of the networks’ layers with a multi-layered perceptron so that we could efficiently use small-sized input data to train the networks. We set one hidden layer for the discriminator and one hidden layer for the generator. Figure 5 illustrates the structure of the GAN. The hidden units are set by 128 units and activated by a leaky rectified linear unit (*Leaky ReLU*), the output units of the generator are activated by the hyperbolic tangent function (*tanh*), and the output unit of the discriminator is activated by the logistic sigmoid function.

The architecture of the GAN is based on the differentiable generator networks [27,30], and the associated loss and cost functions can be defined with a payoff function V, and the default choice for V is
(3)$$V(\theta^{\left(g\right)},\;\theta^{\left(d\right)})=E_{x∼p_{\mathrm{data}}}\log\;d\left(x\right)+E_{z∼p_{\mathrm{model}}}\log\;\left(1-d\left(g\left(z\right)\right)\right),$$
where the generator g transforms the latent variables z into fake samples $g\left(z\right)$ with the associate neural network’s weight parameter $\theta^{\left(g\right)}$, and the discriminator d outputs the payoff values for input x and $g\left(z\right)$ to differentiate the fake samples using the weight parameter $\theta^{\left(d\right)}$. Through the zero-sum game of GAN, we train the networks to solve the min–max problem:(4)$$\underset{\;\;\;\;\;\;\;\;\;\;\;\;\;\;\;g(z;\theta^{\left(g\right)})\;d(x;\theta^{\left(d\right)})}{\arg\;\min\;\max}V\left(g,d\right)$$

In training the generator, Equation (4) leads to a solution of the minimization problem:(5)$$\underset{\;\;\;g\left(z;\theta^{\left(g\right)}\right)\;\;\;}{\arg\;\min\;}E_{z∼p_{\mathrm{model}}}\log\;\left(1-d\left(g\left(z\right)\right)\right)$$

In our study, we aim to generate samples similar to the outliers of the transform to allow the discriminator to experience the differentiation of outlier-like samples, and we suggest that, if the generator produces samples similar to real samples, then the discriminator will suffer in learning knowledge about the outliers. To this end, we apply a loss function to train the generator with lower slopes of the gradient than those used in Equation (5) and, as one of several choices, we use $-\log\;d\left(g\left(z\right)\right)$ instead of $\log\;\left(1-d\left(g\left(z\right)\right)\right)$; looking at Figure 6, which presents the comparison plots for $-\log\;d\left(g\left(z\right)\right)$ and $\log\;\left(1-d\left(g\left(z\right)\right)\right)$, as the value $d\left(g\left(z\right)\right)$ approaches 1, the slope of the gradient in $\log\;\left(1-d\left(g\left(z\right)\right)\right)$ becomes steeper than $-\log\;d\left(g\left(z\right)\right)$, and we suppose that, if we set the generator’s loss function to $-\log\;d\left(g\left(z\right)\right)$, then the discriminator’s response to determine the differentiation is more sensitive than the generator’s sensitiveness to shape the fakes, which will give the discriminator more chances to learn lots of information about the outlier-like samples. Note that if $d\left(g\left(z\right)\right)$ appears near to be 0, it is not necessary to have the discriminator speed up to discern the fakes, because the input is the fake. From that suggestion, in training the generator in our study, we solve the minimization problem:(6)$$\underset{\;\;\;g\left(z;\theta^{\left(g\right)}\right)\;\;\;}{\arg\;\min\;}E_{z∼p_{\mathrm{model}}}-\log\;d\left(g\left(z\right)\right)$$

Now, for the data normalization used to train D-GAN, we divide the input samples by the element-wise maximum absolute values of the training data, and we call this max-abs normalization; i.e., for a given sample transformation T, we normalize it such that
(7)$$T_{scaled}=\frac{T}{T_{max}}$$
where $T_{max}$ is the element-wise maximum absolute values of the training data. The generated samples emitted from the generator are allowed to have a negative (−) sign because of the *tanh* activation followed by the *Leaky ReLu* activation, and we have not constrained the input data to be non-negative values. However, in the inference step for test datasets, we consider a variation treatment that makes it so that positively dominated numbers remain positive, while negatively dominated numbers or positively small numbers are transferred into negative values. This comes from an intuition that, if a falsely estimated transform is given by RANSAC, then the tricky variation treatment may deform the topography of the sample space to make the false sample look falser from the perspective of the G-GAN. For the topography deformation, we consider the following parameter-gauging linear transformation: for a given sample T,
(8)$$T_{deformed}=\xi \left(T-\zeta \right)$$
where ξ and ζ are the hyper-parameters we have to set. In our study, we set ξ=1/2 and ζ=1/2. For a topography deformation technique, the readers may refer to [31].

Next, for the correction of RANSAC’s rejected hypothesis, we use OCICI [4] as one of the refined outlier rejection schemes (see Figure 2). This considers the geometric congruence made by the initial three pairs of the correspondences to measure the transform: for a given set of the initial pairs $\hat{x}$ in the reference image and $\hat{y}$ in the sensed image of the correspondences, (i) we consider the similarity of the triangles ΔABC and ΔA'B'C' that were, respectively, formed by $\hat{x}$ and $\hat{y}$; (ii) we investigate the invariance of a corner’s angle in the sensed image by transforming it with the initially estimated transform into the reference image. Summing up those two constraint factors in the process of OCICI, the authors in [4] defined two loss functions—Θ and Κ—and their associate hyperparameters α and ρ, thus leading them to the following solution for the minimization problem:(9)$$\arg\underset{\hat{x},\;\hat{y}}{\min}\;\alpha \Theta \left(\hat{x},\hat{y}\right)+\rho Κ\left(\hat{x},\hat{y}\right)$$

For the choice for the hyperparameters α and ρ, in our study, we set α=1 and ρ=1 as the default choice, as recommended in [4].

### 2.3. Generative Comparison Network

In this subsection, we introduce a scheme to apply the generator of the GAN (G-GAN) to discover false discriminations of D-GAN and overcome the limitations in the false positive reduction.

As the training of the GAN allows the generator to produce samples that are indistinguishable from real ones, the distribution of the generated samples may resemble that of the real ones. Using this insight, we can search for a generated sample that has the nearest distance to the given hypothesis by measuring the cosine distances over the arbitrary samples generated from the latent space of z. For example, let us assume that there are 1000 latent vectors arbitrarily chosen in the standard Gaussian noise distribution. Then, we generate 1000 samples dependent on the latent vectors and measure, for each of them, the cosine distances from the given hypothesis. Discerning the closest one among the samples, we then verify the true positiveness of the given hypothesis.

In Figure 7, we illustrate the scheme of G-GAN’s inference to measure the cosine distance to RANSAC’s hypothesis. Looking at Figure 7, in the process of G-GAN’s sample generation to measure the cosine distance, we use the *logit* of the output layer in G-GAN to represent the generated sample that comes before the activation of *tanh* and, by implementing this, we intend to increase the numerical variation in the topography of the generated samples such that the values in the generated samples are mapped to the range around (−5, 5), which comes from the fact that the *tanh* goes, approximately, from −1 to +1 for the input going from under −5 to over +5; the generated samples may not need to be evaluated into the range [−1, 1], because we only measure the directional similarity, i.e., cosine distance, of the vectors representing the randomly generated samples. The idea to use the *logit* rather than the activated original output has come from our preliminary experiments. For any given *T* in the training data consisting of 16,378 samples, we generate 1000 samples (the outputs activated by *tanh*), and with the samples’ *logit*s, we organize another 1000 samples. After gathering those generated samples, we measure the distances from *T*. Table 1 lists the queries’ results that can be used to determine if the distances from the *tanh* outputs or their *logit*s are closer to the given *T*. For a given *T*, we query from which group the closest one comes. If the closest one comes from the *logit* group, then we identify it with an inequality such that *C(Logit, T) > C(tanh, T)*, and for the other case, we count it into *C(Logit, T) < C(tanh, T)*. Looking at Table 1, we suggest that the *logit* samples better describe the real samples with closer distances.

For a more specific analysis, we apply the proper orthogonal decomposition (POD) [32] for the sample space of the training data. To obtain the POD bases, we apply the singular vector decomposition [32] for the snapshot matrix consisting of a 9 × 16,378 matrix to produce six POD bases. With the three POD bases that correspond to the three largest eigenvalues, we represent the samples in the form of three-dimensional POD-based coordinates; then, for the three-dimensional represented samples’ space, we conduct the same comparison tests as we did to obtain the results in Table 1. Table 2 lists the results for those queries; looking at Table 2, the POD-based samples’ representations confirm that the *logit* samples move closer to the distribution of the real samples, as can also be seen in Table 1. In Figure 8, we plot the samples’ POD-based remaining three-dimensional representations into the three-dimensional axis with the *XY*-axis (top-left), *XZ*-axis (top-right), and *YZ*-axis (bottom-left). Looking at Figure 8, for the coordinates of the three remaining POD bases, the real samples’ POD-based distribution is dispersed around the origin (0,0); the *tanh* samples’ coordinates are located at the points furthest away from the origin; i.e., the *tanh* samples’ distribution error is much more disturbing than that of the *logit* samples. Note that, as the sample space is organized by 3 × 3 matrices, a three-dimensional vector space shall optimally represent the sample space and, hence, the remaining three POD bases’ representations should have the distribution centered at zero.

Figure 9 illustrates the comparison of a hypothesis given by RANSAC, the closest real sample, and the closest generated sample by G-GAN, where the cosine distance from the hypothesis = 0.65 (D-GAN’s output = 1). In this case, we know that the hypothesis is good and that the real one achieves the best match. Even though the generated one has a relatively far distance, we can employ a threshold $\delta_{G}$ to discern the hypothesis’ validity; i.e., for a given hypothesis $t_{h}$ and the closest generated sample $t_{s}$, if it holds that
(10)$$C(\vec{t_{h}},\vec{t_{s}})>\delta_{G}$$

Then, we assume that hypothesis $t_{h}$ is a true positive, where $\vec{t_{h}}$ and $\vec{t_{s}}$ are the vector forms of $t_{h}$ and $t_{s}$. For the false hypothesis, as shown in Figure 10 and Figure 11, we present two lowly evaluated generated samples; the generated samples’ evaluations are −0.04 and 0.47, whereas the real samples are relatively higher values of 0.84 and 0.53, respectively. Note that the D-GAN’s output is 1 for the case of the falsely estimated hypothesis given in Figure 11. From those preliminary tests, we suggest that, if every generated sample lies on almost lower ranges of the evaluations than the real samples, as can be seen in Figure 10 and Figure 11, then the G-GAN may be useful for querying the detection of the false positiveness made by D-GAN. From this point of view, we modify the model of ODNet to have an additional outlier-searching network with G-GAN so that we introduce the generative comparison network (GCNet) as illustrated in Figure 12. Starting from RANSAC, (i) we discriminate the RANSAC hypothesis using D-GAN, and (ii) if that is discerned as true, (iii) it recalls G-GAN to generate samples to compare with; then, (iv) if it finds a similar one among the G-GAN samples, we finally assume that the RANSAC hypothesis is true; if the RANSAC hypothesis is discerned to be false by D-GAN, or if we find no similar one among the G-GAN samples, then we recall OCICI.

## 3. Results and Analysis

### 3.1. Experimental Result Applying ODNet

Focusing on the estimation of the transform created by applying ODNet, we test certain pairs of images for which RANSAC fails to reject the outliers. To evaluate the output value of D-GAN into true or false, for a given input sample x, we set a threshold $\delta_{D}$, such that if it holds
(11)$$d\left(x\right)>\delta_{D}$$

Then, sample x is determined as a true one. For the experiments, we set $\delta_{D}=0.4$, which is given as one of our empirical choices. Note that, among the pairs of images that have been used to extract training samples, for one pairing, RANSAC has rendered estimated operators whose discriminator outputs vary sensitively. However, most of the outputs of the discriminator are given as 1 or 0; therefore, in our experiments, the threshold *δD* has given less effects to the numerical results for ODNet. In Table 3, we present a preliminary result obtained using the discriminator of the GAN for the test images that are not included in the training process of the GAN. For each pairing, we attempt 20 trials of full estimations, and each trial case is set differently for the given ϵ defined in Equation (2). We count good transforms as true positive ones and the outliers that we can heuristically judge to be false ones as negative ones. As presented in Table 3, the total false positive rate is reduced by D-GAN; however, we also find that, as the false negative cases occur, then the unnecessary extra calculation cost to recall OCICI is added for ODNet.

Table 4 presents the comparison results for the calculation cost and accuracy for the datasets set by ϵ=1.1, 1.2,⋯, 1.5, which have been obtained using RANSAC, ODNet, and OCICI. Looking at Table 4, for ϵ=1.1, 1.2, ODNet and OCICI produce 100% accuracy; note that the maximum value in the parentheses indicates the accuracy accomplished by ODNet if the rejected RANSAC’s correction method is replaced by an ideal one instead of OCICI; in total, we see that ODNet outperforms the others in terms of accuracy and reduces the cost consumed for the OCICI scheme. In Table 5, we give the false positive cases (FPs) versus whole pairings and the false negative cases (FNs) among the negatively discriminated ones by D-GAN, and the true negative detection (TND) that denotes the cases in which the D-GAN accurately discriminates the RANSAC’s falsely estimated cases.

### 3.2. Experimental Result Applying GCNet

ODNet helps to reduce the number of falsely estimated transforms produced by RANSAC; however, it could not reduce the number of FPs to zero, because the D-GAN failed to perfectly detect the outliers for several values of ϵ.

At this point, we have the test for GCNet to reduce the rate of FPs. For the threshold $\delta_{G}$ with which to determine the distance between two samples in Equation (10), we set $\delta_{G}=0.5$. Table 6 presents the results for comparison with RANSAC and OCICI for the test datasets organized by the descriptors’ correspondences with ϵ=1.3, 1.4, 1.5. Table 7 gives the related results regarding the FPs, FNs, and TND of GCNet. Looking at Table 6 and Table 7, we can see that GCNet outperforms the other methods by relatively reducing the calculation costs and enhancing the accuracy; the FP rate is zero in all cases, i.e., the FP rate made by RANSAC can be perfectly detected by GCNet and, hence, a GCNet-based method followed by an ad hoc inlier selection method superior to OCICI may set the best conditions to produce a true hypothesis. Figure 13 shows a bar graph depicting the numerical properties of ODNet and GCNet to summarize the efficiency in reducing the FP rate that occurs with RANSAC. For calculation costs, ODNet and GCNet are competitive, whereas for the FP reduction rate, GCNet outperforms ODNet; further, for our test datasets of the drone-based aerial images, our proposed method’s model of the neural network is experimentally well verified.

### 3.3. Discussion and Experiments for Some Miscellaneous Images

Through the experiments described above, we have verified the performances of the two proposed neural network-based methods, ODNet and GCNet, and by applying the methods, we have reduced the number of falsely estimated hypotheses of the transforms for the correspondences of the given descriptors. The operation of the proposed methods is dependent on the quality of the trained neural networks. We have trained them with the dataset of the drone-based aerial images introduced in [4], and we resized the images into the resolution of 200 × 250 for use in our experiments; i.e., the trained neural networks would be dependent on the given images’ resolution scale. In other words, the proposed methods may function invariantly for images that have the same designated resolutions, regardless of the objects’ kinds of views in the images. Note that the neural networks of the proposed methods have been trained for the sample transforms that have a resolution of 3 × 3; therefore, the proposed methods’ computational costs would be substantially lower than those of any of the other developed deep learning-based methodologies.

To compare the methods’ performances, we found out that the GCNet’s employment of G-GAN is useful and that it outperforms the D-GAN-based ODNet in detecting the outliers. Even though GCNet emitted relatively many false rejections of the true hypothesis compared to ODNet, as the calculation costs of both methods are competitive, the trade-off effect may be negligible, at least for our ad hoc datasets (see Figure 13). For the details of the performance comparison, we visualize the cases that the methods have detected for RANSAC’s FPs for the test datasets in Figure 14. Meanwhile, Figure 15 presents the cases in which D-GAN fails to detect the FPs and the cases that G-GAN detects for the FPs with small discrepancies. From Figure 14 and Figure 15, we can see that the G-GAN has recognized FPs in better detail and more diversely.

As an issue regarding the proposed methods, we discuss how the use of the methods may be efficient. The methods are modeled to recognize the FP cases that are falsely estimated by RANSAC. The G-GAN functions as the confirming network while using its generated samples that are randomly produced to be compared for the distances from the RANSAC’s given hypothesis. Looking at Figure 9, Figure 10 and Figure 11, real samples are found to compare the possibility of the generated sample’s appearances in the trained generator’s output distribution. If the real sample’s distribution contains any true recognition of whether the RANSAC’s hypothesis is an FP or TP, then this would mean that the proposed neural network’s training model could only be used to compress the real samples’ (training data) information into the neural network’s memory. In Figure 16, we present a case wherein a real sample is found to assert that the given RANSAC’s hypothesis can be found in the true hypothesis distribution. However, the hypothesis makes a strong case for itself to be an FP, and the G-GAN discerns it as one of the FPs.

Figure 17 presents (a) the types of the stitching outputs mapped by the sample transforms that were used as the GAN’s training data, and (b) a miscellaneous stitching output that is not included in the output types for GAN’s training data, where we set the threshold ϵ to be 1.2. For this example, the G-GAN outputs 0.56 for the hypothesis’s distance, and if we set the threshold $\delta_{D}=0.5$, then it shall be discerned as a true one. Table 8 lists the results of an experiment that we have conducted to rigorously verify the robustness of the proposed methods in many severe conditions. For the test, we have performed 20 hypothesis trials using RANSAC, which has been constrained to operate for only three epochs for inference, which we have implemented to increase the probability of FPs to appear, where the thresholds $\delta_{D}$ and $\delta_{G}$ are set as 0.4 and 0.5 for D-GAN and G-GAN, respectively. In this case, the G-GAN also outperforms D-GAN in discerning outliers, even though the G-GAN has made one misdetection among the 40 trials in total. For the case in which G-GAN was mistaken, we suggest that the situation may happen in which the given images’ correct mapping is out of the types of the forms that the images employed in the training dataset have; i.e., the generated sample of G-GAN is unfamiliar for the network of G-GAN itself. For this issue, we may have to consider better training strategies to overcome the networks’ suffering for the unfamiliar styles of the test samples and enhance the performances of the proposed models to operate in severe conditions that affect the descriptors.

## 4. Conclusions

Recently, advancements in AI have led to the development of state-of-the-art techniques in computer science. The development of an outlier rejection scheme for correspondence problems represents an important issue in the current area of computer vision. Although deep learning-based technologies for the correspondence problems, and for the stitching problems in particular, have been actively developed with various models, it was difficult to find a state-of-the-art direct application of deep learning to reduce the falsely estimated transform’s hypothesis produced by some baseline methods, such as RANSAC. In this study, we proposed a deep learning-based outlier rejection scheme by applying the architecture of GAN. In applying the discriminator and the generator of the GAN model, we proposed the outlier rejection networks called ODNet and GCNet. We have tested the methods for drone-based aerial images and some miscellaneous images. The methods operated with outstanding performances, reducing the FPs that RANSAC has produced due to its inherent computational limitations. GCNet functioned with relatively stable and noble performances, even for receiver-operated tough conditions, and it outperformed ODNet in reducing FPs.

Regarding the limitations of the present work that provide directions for future research, we suggest that there may be a need for training methodologies that can enhance the neural networks’ performances when inferring recognitions, in addition to the ad hoc topographical deformation techniques; for example, we would set a loss function to train GAN, which aims to differentiate the adversarial networks’ training speeds so that we deliberately handle each network’s training status for the enhancement of the networks’ cognition abilities. Further, the organization of the training samples can also be improved upon, and we have to apply the methods with images of various formats, resolutions, volumes, etc.

## Figures and Tables

**Figure 1 sensors-22-02474-f001:**
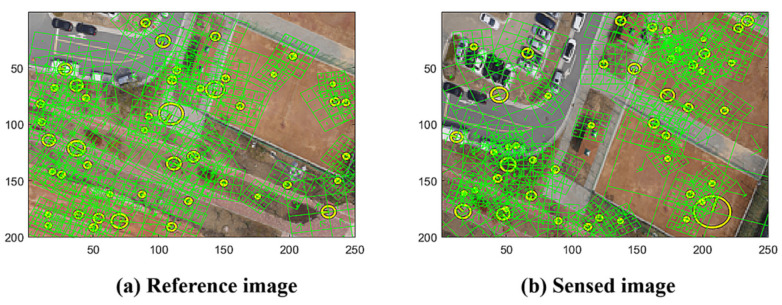
Reference (**a**) and sensed (**b**) images with SIFT.

**Figure 2 sensors-22-02474-f002:**
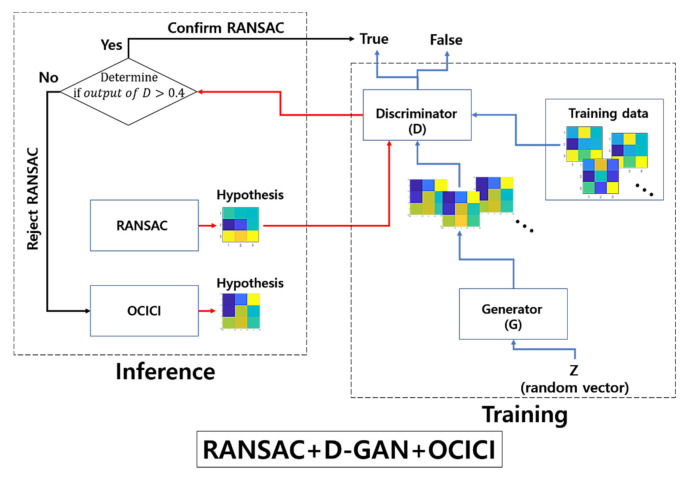
Architecture of the outlier discrimination network consisting of RANSAC + D-GAN + OCICI.

**Figure 3 sensors-22-02474-f003:**
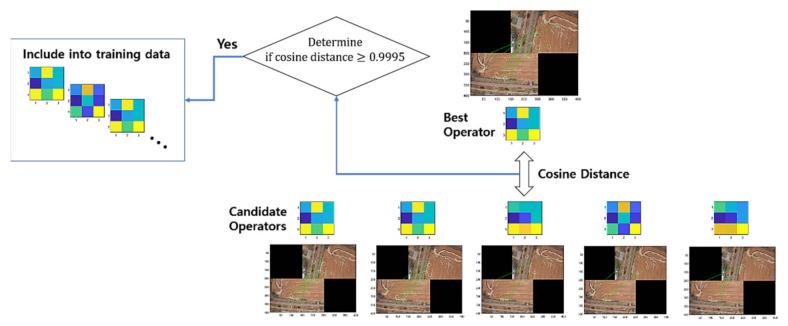
Acquisition of the training data.

**Figure 4 sensors-22-02474-f004:**
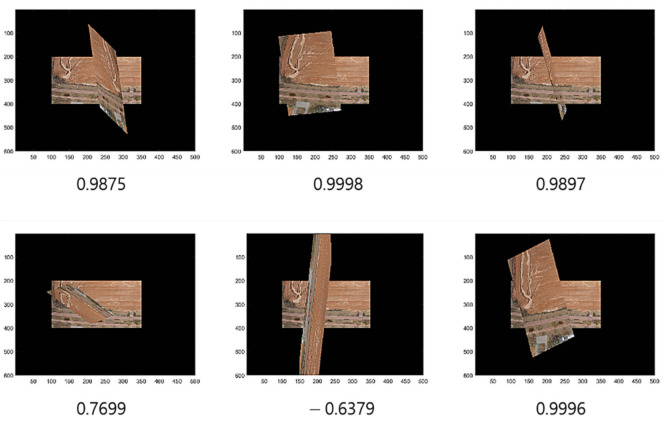
Cosine distances and overlapped images.

**Figure 5 sensors-22-02474-f005:**
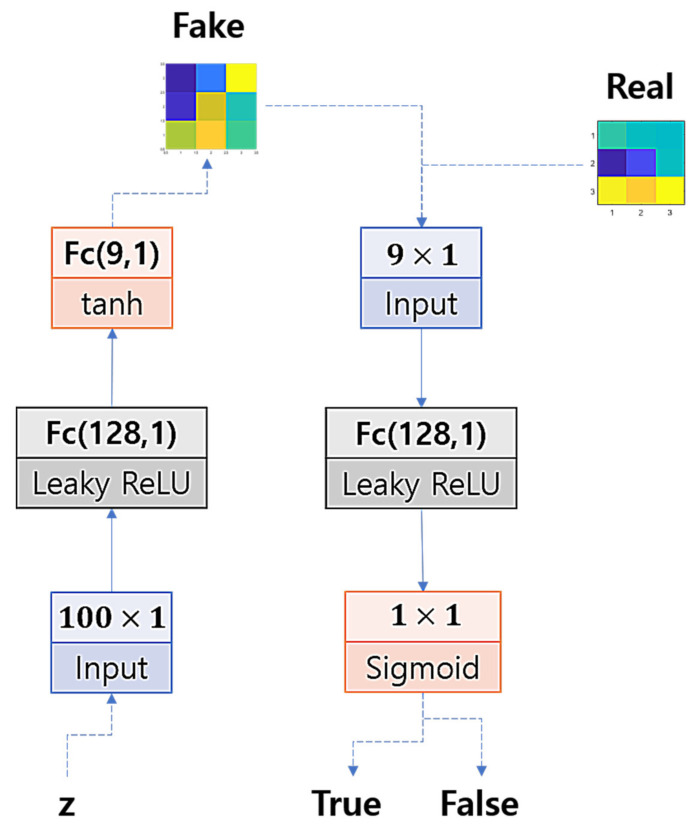
GAN architecture.

**Figure 6 sensors-22-02474-f006:**
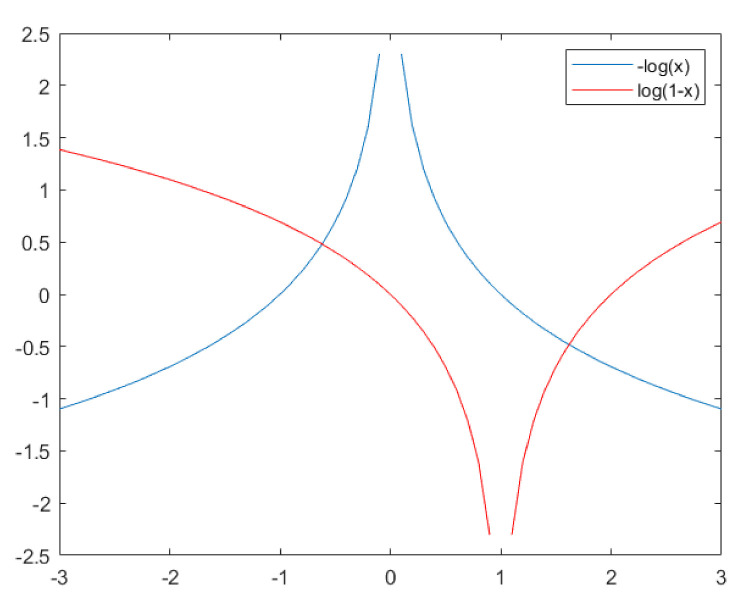
Comparison of the slopes of $-\log\left(x\right)$ and $\log\left(1-x\right)$.

**Figure 7 sensors-22-02474-f007:**
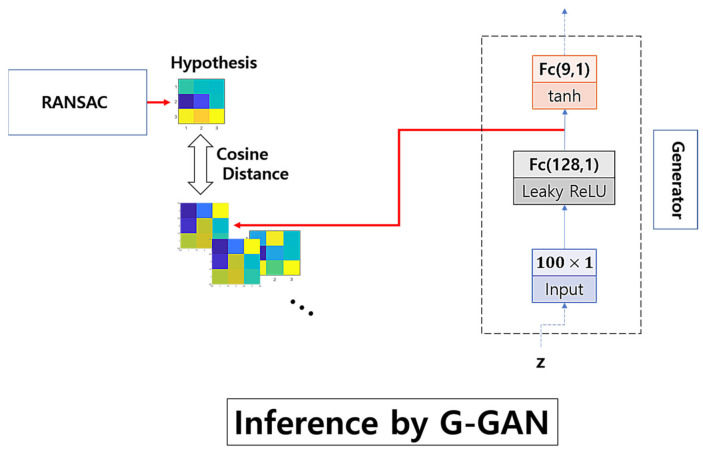
Scheme of G-GAN’s inference.

**Figure 8 sensors-22-02474-f008:**
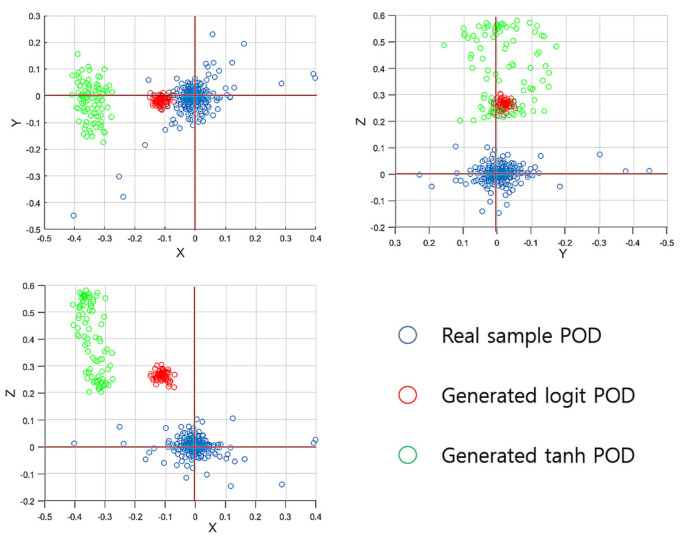
Plots of the samples’ POD-represented vectors with the remaining three-dimensional minimal POD bases.

**Figure 9 sensors-22-02474-f009:**
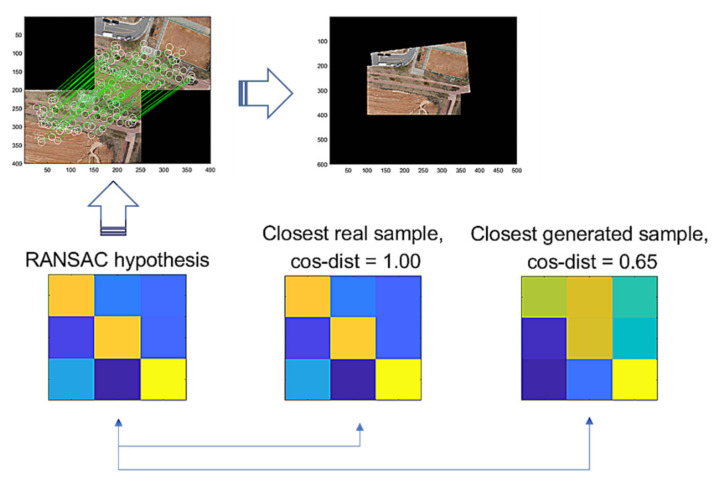
RANSAC’s hypothesis and its closest sample in the training dataset of GAN, and the closest generated sample.

**Figure 10 sensors-22-02474-f010:**
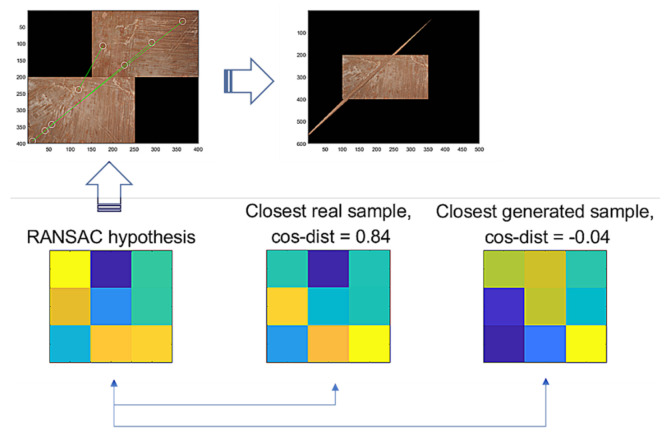
RANSAC’s hypothesis and its closest sample in the training dataset of GAN, and the closest generated sample.

**Figure 11 sensors-22-02474-f011:**
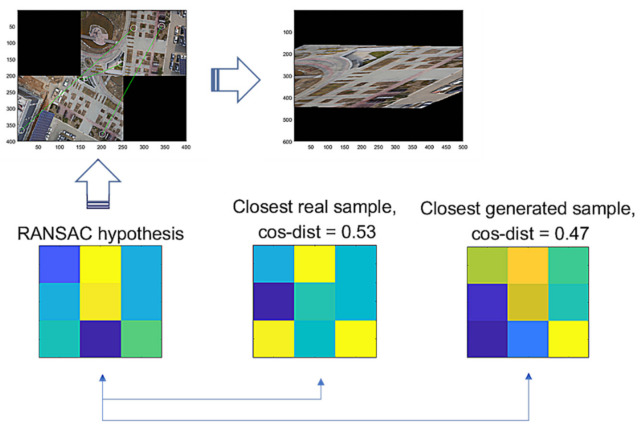
RANSAC’s hypothesis and its closest sample in the training dataset of GAN, and the closest generated sample.

**Figure 12 sensors-22-02474-f012:**
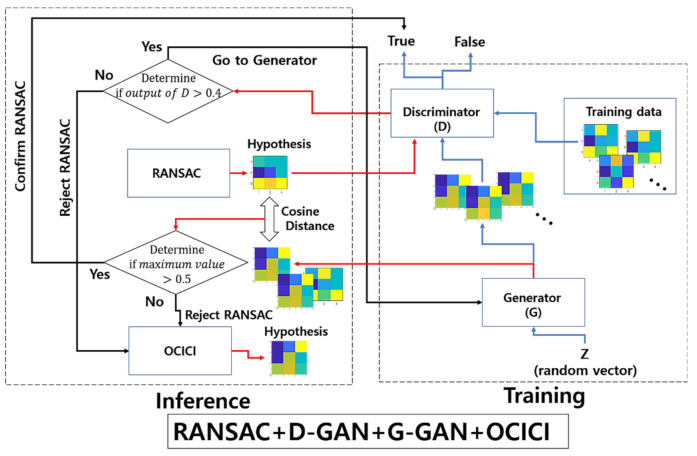
Scheme of generative comparison network consisting of RANSAC + D-GAN + G-GAN + OCICI.

**Figure 13 sensors-22-02474-f013:**
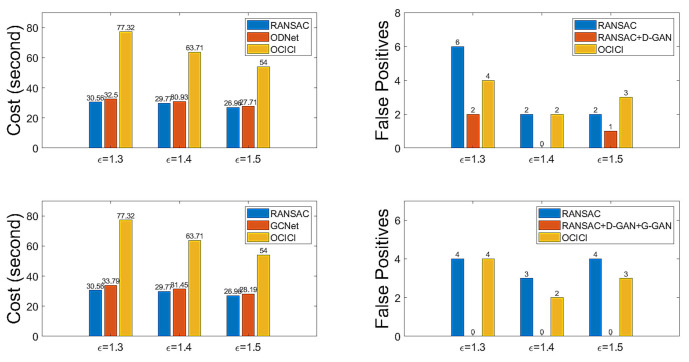
Discrimination performance (cost and number of false positive hypothesis mappings) for the whole dataset for ϵ=1.3, 1.4, 1.5.

**Figure 14 sensors-22-02474-f014:**
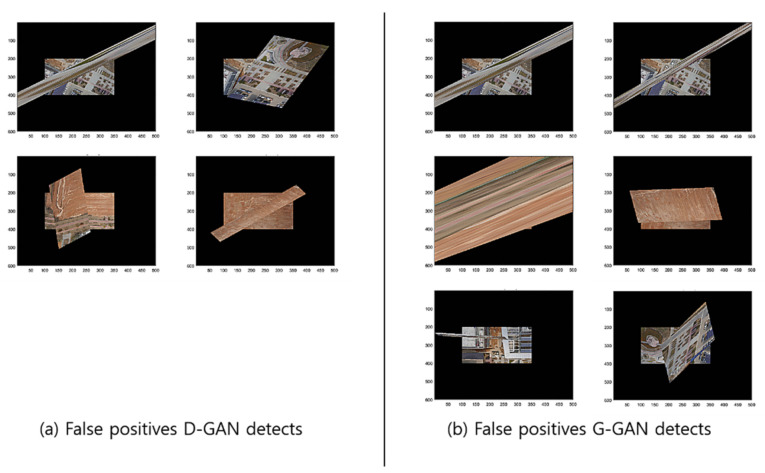
Discrimination types that D-GAN and G-GAN recognize for test datasets.

**Figure 15 sensors-22-02474-f015:**
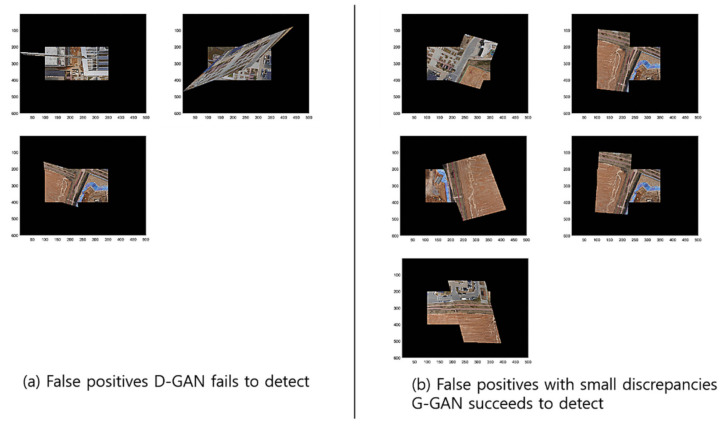
Discrimination types that D-GAN fails to detect and G-GAN succeeds in detecting despite small discrepancies for test datasets.

**Figure 16 sensors-22-02474-f016:**
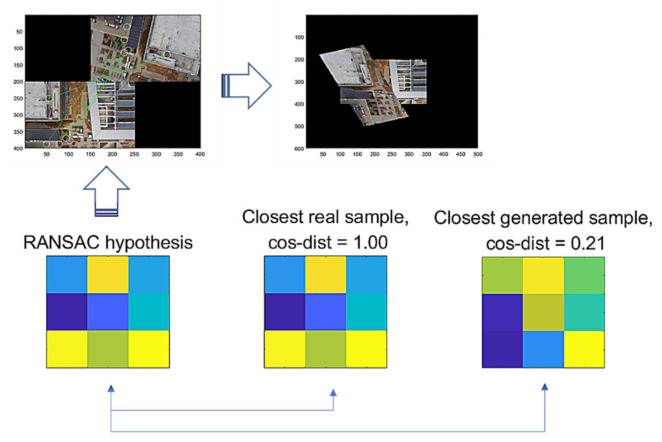
RANSAC’s hypothesis and its closest sample in the training dataset of GAN, and the closest generated sample by G-GAN.

**Figure 17 sensors-22-02474-f017:**
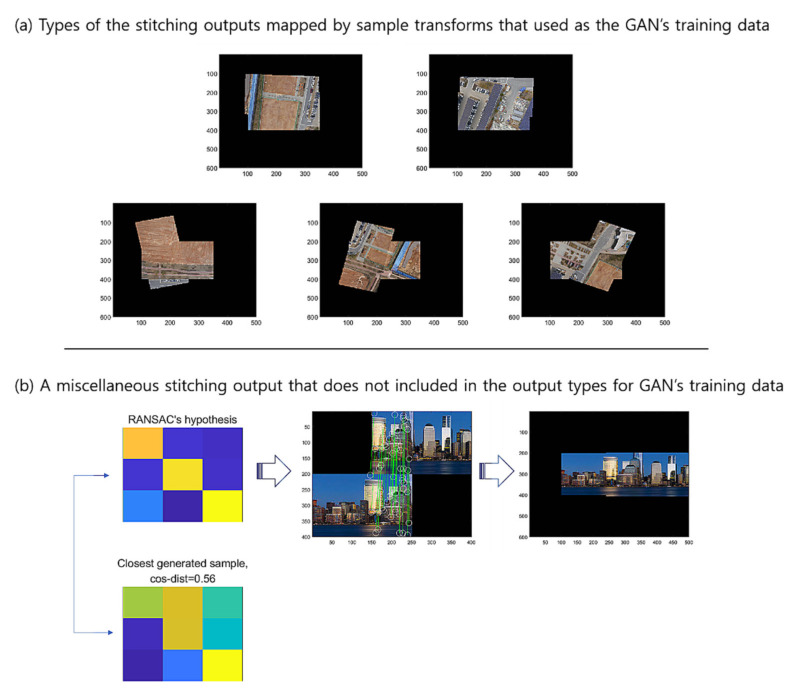
Types of stitching outputs mapped by sample transformations.

**Table 1 sensors-22-02474-t001:** Distance comparison for *logit* and *tanh* to alternatively distinguish the one closer to the real samples in the total 16,378 pieces of training data.

	** *C(Logit, T) > C(tanh, T)* **	** *(Logit, T) < C(tanh, T)* **	**Total**
Number of cases	15,601	777	16,378

**Table 2 sensors-22-02474-t002:** POD version distance comparison for *logit* and *tanh* to alternatively distinguish the one closer to real samples in the total 16,378 pieces of training data, where the associated POD bases consist of the maximal three POD bases among six POD bases.

	*C(Logit, T) > C(tanh, T)*	*(Logit, T) < C(tanh, T)*	Total
Number of cases	15,325	1053	16,378

**Table 3 sensors-22-02474-t003:** Outlier rejection test of RANSAC and D-GAN for several test images.

ϵ	Stitching View by RANSAC	False Positive Case vs. Total Cases by RANSAC	False Positive Cases vs. Total Cases by D-GAN
ϵ=1.1	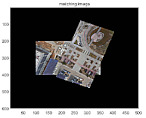	0/20	0/20 (False negative rate = 100%)
ϵ=1.2	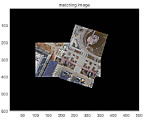	0/20	0/20 (False negative rate = 100%)
ϵ=1.1	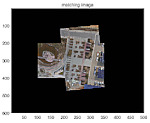	0/20	0/20 (True positive rate = 100%)
ϵ=1.2	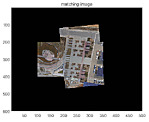	0/20	0/20 (True positive rate = 100%)
ϵ=1.1	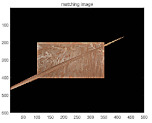	11/20	1/20 (True positive cases = 0)
ϵ=1.2	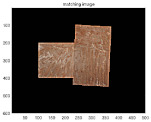	0/20	0/20 (False negative rate = 100%)
Total false positive rate		9.1667%	2.439%

**Table 4 sensors-22-02474-t004:** Cost and accuracy for the datasets set by ϵ=1.1, 1.2,⋯, 1.5, with RANSAC, ODNet, and OCICI.

	ϵ	RANSAC	ODNet	OCICI
Accuracy	ϵ=1.1	96.8254%	100%	100%
	ϵ=1.2	98.4127%	100%	100%
	ϵ=1.3	90.4762%	93.6508% (max 96.8254%)	93.6508%
	ϵ=1.4	96.6102%	98.3051% (max 100%)	96.6102%
	ϵ=1.5	96.5517%	96.5517% (max 98.2759%)	94.8276%
Cost	ϵ=1.1	34.2061	66.6555	318.6295
	ϵ=1.2	89.5633	94.3174	145.6169
	ϵ=1.3	30.5566	32.4998	77.3177
	ϵ=1.4	29.7655	30.9285	63.7132
	ϵ=1.5	26.9586	27.7069	54.0032

**Table 5 sensors-22-02474-t005:** Discrimination accuracy and false positive rate of ODNet for the datasets set by ϵ=1.1, 1.2,⋯, 1.5.

	ϵ=1.1	ϵ=1.2	ϵ=1.3	ϵ=1.4	ϵ=1.5
Accuracy	85.7143%	85.7143%	90.4762%	89.8305%	89.6552%
FP	0/52	0/53	2/55	0/51	1/52
FN	9/11	9/10	4/8	6/8	5/6
TND	2/2	1/1	4/6	2/2	1/2

**Table 6 sensors-22-02474-t006:** Cost and accuracy for the whole dataset for ϵ=1.3, 1.4, 1.5.

	ϵ	RANSAC	GCNet	OCICI
Accuracy	ϵ=1.3	93.6508%	93.6508% (max 100%)	93.6508%
	ϵ=1.4	94.9153%	96.6102% (max 100%)	96.6102%
	ϵ=1.5	93.1034%	94.8276% (max 100%)	94.8276%
Cost	ϵ=1.3	30.5566	33.7912	77.3177
	ϵ=1.4	29.7655	31.4479	63.7132
	ϵ=1.5	26.9586	28.1906	54.0032

**Table 7 sensors-22-02474-t007:** Discrimination accuracy and false positive rate of GCNet for the datasets set by ϵ=1.3, 1.4, 1.5.

	ϵ=1.3	ϵ=1.4	ϵ=1.5
Accuracy	76.1905%	79.6610%	82.7586%
FP	0/44	0/44	0/44
FN	15/19	12/15	10/14
TND	4/4	3/3	4/4

**Table 8 sensors-22-02474-t008:** Results for 20 hypothesis trials by RANSAC, which has been constrained to operate for only three epochs for inference, where the thresholds $\delta_{D}$ and $\delta_{G}$ are set as 0.4 and 0.5 for D-GAN and G-GAN, respectively.

Matching Correspondences	False Stitching Output	FP (for 20 Trials)
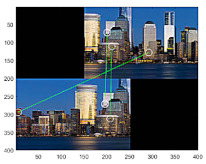 ϵ=1.2	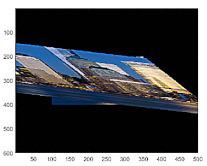	RANSAC: 10%
		D-GAN: 5%
		G-GAN: 5%
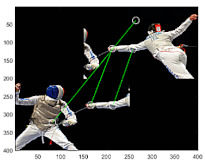 ϵ=1.2	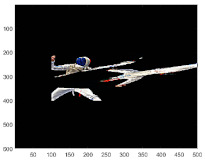	RANSAC: 40%
		D-GAN: 15%
		G-GAN: 0%

## Data Availability

The data presented in this study are available in https://github.com/seojksc/seojk-kuids (accessed on 10 November 2021).

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
