# Peer review of "A Cognitive Sample Consensus Method for the Stitching of Drone-Based Aerial Images Supported by a Generative Adversarial Network for False Positive Reduction"

_sensors, 2022, doi:10.3390/s22072474_

Round 1

Reviewer 1 Report

This paper presents  deep-learning-based outlier rejection schemes applying the architecture of GAN. 

There are several issues to be addressed.

1. Authors mentioned "drone-based aerial images" in the title of this paper. 
However, it seems that the proposed method is about image outlier reduction,  which can be applied to general images, not specific for drone-based aerial images. 
Does the proposed methods contains any specific features customized to drone-based aerial images?  
If so,  please present the customized factors clearly and in more detail.

2. On page 8, author mentioned "we set ?? = 0.4 which is given as one of our empirical choices.". 
It would be helpful that authors present more detailed justification about 
the value of ?? set in this paper. 

3. Some sentences throughout the paper are way too long so it is hard to understand them.
For example, the following sentence on Page 7 is about 9 lines long. 
"For that purpose, we apply a loss function for the training of the generator with lower slopes of the gradient than that used in Eq.(5), and ..... discriminator have more chances to learn lots of information of the outlier-like samples." 

4. In this paper, authors put methodologies and experimental results into a single section, Section 2. 
It would be helpful to divide it into two separate sections, one for the proposed methodologies and one for results and analysis. 

Reviewer 2 Report

  1. There are the same statement “Recently, the development of artificial intelligence leads the state-of-the-art techniques in computer sciences. In the area of computer vision, the development of the outlier rejection scheme for correspondence problems is important.”both in Abstract and Conclusions. Please delete one and narrow down your Abstract.
  2. Titles of the sections don’t need text-indent, such as Section 2.1 Extraction of local descriptors and geometric correspondence, and the font size of them should be smaller than the previous level titles, such as Section 2 Method Description.
  3. The name of your Figures should be abridge, such as the name of Figure 1 should be changed to Reference (a) and sensed (b) images with SIFT descriptors, and the redundant explanations should be put in the upper paragraph.
  4. There are many format errors in the paper, such as the row numbers overlap with the Table 2, 4, 5, and 7, and many sentences wrap in the middle of the word, please check and revise
  5. I find most of the problems are due to writing. I suggest the author polish the writing so the readers can understand better.

Reviewer 3 Report

The paper needs to be revised for better clarity.

(1) In Figure2 and Figure3,  it shows many 3-by-3 images/hypothesis. What are these? images? operators (see Figure3)  image patches?  affine transform (I assume this, please confirm) ?

(2) In Figure 2,  "if value > 0.4". what is this "value"? the output from the discriminator?

(3) again, what is the meaning of "hypothesis" in Figure 2?

a possible alignment (i.e., affine transform)? 

(4) The GAN is trained to generate good transforms (3x3 matrix, I assume)

what is the input z? a random vector?

If z is a random vector, then the GAN only servers as a method to generate all possible transforms.

(5) By using the discriminator of GAN, only the hypothesis similar to those in the training data will be accepted. If the training data has all possible transforms, then the discriminator will become useless because any hypothesis will be accepted.

The performance of the method is dependent on the similarity between the transforms in the training data and the transforms in the testing data, which limits the practical use of this method

Round 2

Reviewer 1 Report

Authors reflects most of  reviewers' concerns in the paper. 
It would be better to include "authors'response 1" into the paper, not just in the response. 

Author Response

Minor revision

Reviewer 1 : We present special thanks for the reviewer’s rigorous effort to review our manuscript. We have the point-by-point response to the reviewer’s comments as follows

Comments and Suggestions for Authors: 
Authors reflects most of  reviewers' concerns in the paper. 
It would be better to include "authors'response 1" into the paper, not just in the response. 

Response: 
We have revised the manuscript according to the reviewer's comment and, now, we have checked that the contents in the response 1 are well included in Line 96-114.

Reviewer 2 Report

Thank the authors for their efforts. The authors have adequately addressed all my concerns in the review, and did a good job to revise and improve the paper. The paper now is suitable for publication in Sensors in its current form.

Author Response

We present special thanks for the reviewer’s rigorous effort to review our manuscript.

Comments and Suggestions for Authors:
Thank the authors for their efforts. The authors have adequately addressed all my concerns in the review, and did a good job to revise and improve the paper. The paper now is suitable for publication in Sensors in its current form.